# Contrastive Consistent Representation Distillation

## Abstract

The combination of knowledge distillation with contrastive learning has great potential to distill structural knowledge. Most of the contrastive-learning-based distillation methods treat the entire training dataset as the memory bank and maintain two memory banks, one for the student and one for the teacher. Besides, the representations in the two memory banks are updated in a momentum manner, leading to representation inconsistency. In this work, we propose **Co**ntrastive **Co**nsistent **R**epresentation **D**istillation (**CoCoRD**) to provide consistent representations for efficient contrastive-learning-based distillation. Instead of momentum-updating the cached representations, CoCoRD updates the encoders in a momentum manner. Specifically, the teacher is equipped with a momentum-updated projection head to generate consistent representations. The teacher representations are cached in a fixed-size queue which serves as the only memory bank in CoCoRD and is significantly smaller than the entire training dataset. Additionally, a slow-moving student, implemented as a momentum-based moving average of the student, is built to facilitate contrastive learning. CoCoRD, which utilizes only one memory bank and much fewer negative keys, provides highly competitive results under typical teacher-student settings. On ImageNet, CoCoRD-distilled ResNet50 outperforms the teacher ResNet101 by 0.2% top-1 accuracy. Furthermore, in PASCAL VOC and COCO detection, the detectors whose backbones are initialized by CoCoRD-distilled models exhibit considerable performance improvements.

## 1 Introduction

The remarkable performance of convolutional neural networks (CNNs) in various computer vision tasks, such as image recognition (He et al., 2016; Huang et al., 2017) and object detection (Girshick, 2015; Ren et al., 2015; Redmon & Farhadi, 2017), has triggered interest in employing these powerful models beyond benchmark datasets. However, the cutting-edge performance of CNNs is always accompanied by substantial computational costs and storage consumption. Early study has suggested that shallow feedforward networks can approximate arbitrary functions (Hornik et al., 1989). Numerous endeavors have been made to reduce computational overheads and storage burdens. Among those endeavors, Knowledge Distillation, a widely discussed topic, presents a potential solution by training a compact *student* model with knowledge provided by a cumbersome but well-trained *teacher* model.

The majority of distillation methods induce the student to imitate the teacher representations (Zagoruyko & Komodakis, 2017; Park et al., 2019; Tian et al., 2020; Hinton et al., 2015; Chen et al., 2021b;c; Yim et al., 2017; Tung & Mori, 2019; Ahn et al., 2019). Although representations provide more learning information, the difficulty of defining appropriate metrics to align the student representations to the teacher ones challenges the distillation performance. Besides, failing to capture the dependencies between representation dimensions results in lame performance. To enhance performance, researchers attempt to distill structural knowledge by establishing connections between knowledge distillation and contrastive learning (Tian et al., 2020; Chen et al., 2021b).

To efficiently retrieve representations of negative samples for contrastive learning, memory banks cache representations which are updated in a momentum manner, as shown in Fig. 1. However, the student is optimized sharply by the training optimizer. The student representations in the memory bank are inconsistent because the updated representations differ from those not updated in that iteration. Therefore, the student can easily contrast the positive and negative samples, keeping the student from learning good features. The storage size of memory bank is another factor of concern when applying contrastive-learning-based distillation methods. As in (Tian et al., 2020; Chen et al.,

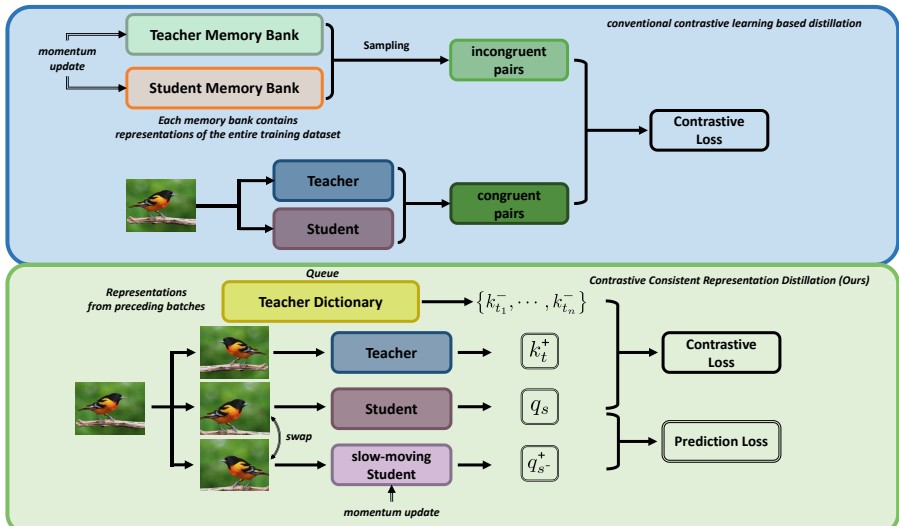

Figure 1: The general pipelines of contrastive learning based knowledge distillation methods and CoCoRD. Instead of momentum updating the representations, CoCoRD updates the encoder in a momentum manner. The teacher dictionary which contains representations from preceding batches is implemented as a queue.

2021b), there are two memory banks and each of them contains representations of all training images, leading to massive GPU memory usage on large-scale datasets.

Motivated by the discussion above, we propose **Co**ntrastive **Co**nsistent **R**epresentation **D**istillation (CoCoRD) as a novel way of distilling consistent representations with one fixed-size memory bank. Specifically, CoCoRD is composed of four major components, as shown in Fig. 2: (1) a fixed-size queue which is referred to as the teacher dictionary, (2) a teacher, (3) a student, and (4) a slow-moving student. From a perspective of considering contrastive learning as a dictionary look-up task, the teacher dictionary is regarded as the memory bank, where all the representations serve as the negative keys. The encoded representations of the current batch from the teacher are enqueued. Once the queue is full of representations, the oldest ones are dequeued. By introducing a queue, the size of the memory bank is decoupled from dataset size and batch size, allowing it to be considerably smaller than dataset size and larger than the commonly-used batch size. The student is followed by a projection head, which maps the student features to a representation space. The teacher projection head is initialized the same as the student one and is a momentum moving average of the student projection head if the teacher and the student have the same feature dimension; otherwise, the teacher projection head is randomly initialized and not updated. Since the contrast through the teacher dictionary is to draw distinctions on an instance level, the cached teacher representations which share the same class label as the student ones are mistakenly treated as negative keys, resulting in noise in the dictionary. To alleviate the impact of the noise, a slow-moving student, implemented as a momentum moving average of the student, is proposed to pull together anchor representations and class-positive ones. As shown in Fig. 2, with a momentum-updated projection head, the slow-moving student projects a data augmentation version of the anchor image to the representation space, which serves as the instance-negative but class-positive key. The main contributions are listed as follows:

- We utilize only one lightweight memory bank (teacher dictionary), where all the representations are treated as negative keys. We experimentally demonstrate that a miniature teacher dictionary with much fewer negative keys can be sufficient for contrastive learning in knowledge distillation.

- We equip the well-trained teacher with a momentum-updated projection head to provide consistent representations for the teacher dictionary. Besides, a slow-moving student provides class-positive representations to alleviate the impact of the potential noise in the teacher dictionary.

- We verify the effectiveness of CoCoRD by achieving the state-of-the-art performance in 11 out of 13 student-teacher combinations in terms of model compression. On ImageNet, the CoCoRD-distilled ResNet50 can outperform the teacher ResNet101 by 0.2% top-1 accuracy. Moreover, we initialize the backbones in object detection with CoCoRD-distilled weights and observe considerable performance improvements over the counterparts that the vanilla students initialize.

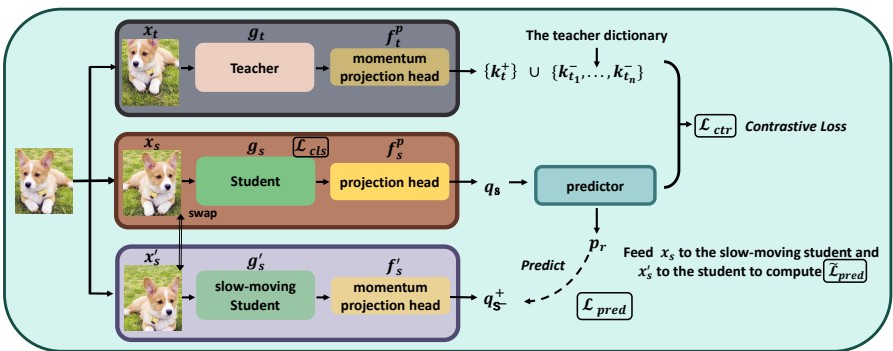

Figure 2: Illustration of the proposed CoCoRD. Note that $q_{s^-}^+$ is detached from the computational graph during the distillation process. $\widetilde{q}_{s^-}^+$, which obtained by feeding $x_s$ to the slow-moving student, is also detached. The teacher is frozen and the teacher dictionary does not receive gradient.

## 2  RELATED WORK

### 2.1  KNOWLEDGE DISTILLATION

Hinton et al. (Hinton et al., 2015) first propose distilling the softened logits from the teacher to the student. After the representative work (Hinton et al., 2015), various knowledge distillation methods (Wang & Li, 2021; Song et al., 2021; Passban et al., 2020; Chen et al., 2021a;c) aim to distill more informative knowledge via intermediate features. Among them, Passban et al. (Passban et al., 2020) fuse all teacher information to avoid the loss of significant knowledge. Chen et al. (Chen et al., 2021a) propose semantic calibration based on the attention mechanism for adaptively assigning cross-layer knowledge. Chen et al. (Chen et al., 2021c) introduce a novel framework via knowledge review in which the knowledge of multiple layers in the teacher can be distilled for supervising one layer of the student. However, the methods mentioned above have difficulty in defining appropriate metrics to measure the distance between the student representations and the counterparts from the teacher. There are a few recent works exploiting the dependencies between representation dimensions based on contrastive learning (Tian et al., 2020; Chen et al., 2021b) for boosting the distillation performance. In particular, Tian et al. (Tian et al., 2020) formulate capturing structural knowledge as contrastive learning and maximize the lower bound of mutual information between the teacher and the student. Chen et al. (Chen et al., 2021b) leverage primal and dual forms of Wasserstein distance, where the dual form yields a contrastive learning objective. In summary, the core of knowledge distillation lies in the definition of knowledge and the way the knowledge is distilled.

### 2.2  CONTRASTIVE LEANING

The main goal of contrastive learning is to learn a representation space where anchor representations stay close to the representations of the positive samples and distant from those of the negative samples. Contrastive learning is a powerful approach in self-supervised learning. To learn powerful feature representations in an unsupervised fashion, Wu et al. (Wu et al., 2018) consider each instance as a distinct class of its own and use noise contrastive estimation (NCE) to tackle the computational challenges. Contrastive learning is first combined with knowledge distillation by CRD (Tian et al., 2020), which aims at exploring structural knowledge. In addition to CRD (Tian et al., 2020), WCoRD (Chen et al., 2021b) combines LCKT (Chen et al., 2021b) and GCKT (Chen et al., 2021b) based on Wasserstein dependency measure in contrastive learning (Ozair et al., 2019). However, the memory banks in CRD and WCoRD contain representations of all the training images, which bring about storage challenges on large-scale datasets. Besides, momentum updates to representations can also lead to inconsistent representations that negatively affect the distillation performance. From a perspective of considering contrastive learning as a dictionary lookup task, we implement the memory bank as a first-in-first-out queue where all included representations serve as negative keys.

## 3  METHOD

The key idea of combining knowledge distillation with contrastive learning is straightforward. With knowledge distillation, a proficient teacher can provide consistent representations that are beneficial for contrastive learning. With contrastive learning, the student can obtain powerful features whose representations are close to the positive teacher representations and distant from the negative ones in a representation space. Contrastive learning can be generally formulated as a dictionary look-up task. Given a query $q$ and a dictionary $K$ with $N$ keys: $K = \{k_1, \cdots, k_N\}$, contrastive learning matches the query $q$ to the positive key $k_+$ and pushes $q$ away from the negative keys cached in $K$.

### 3.1 CONTRAST AS LOOKING UP IN THE TEACHER DICTIONARY

In CoCoRD, the negative keys are encoded by the teacher and cached in a fixed-size queue which is referred to as the teacher dictionary. Given an input image $x$, two views of $x$ under random data augmentations form a positive pair (a query and a positive sample), which is encoded in each iteration.

We define the input to the student $S$ as the query $x_s$ and the input to the teacher $T$ as the positive sample $x_t$. The outputs at the penultimate layer (before the last fully-connected layer) are projected to a representation space by a projection head. For simplicity of notation, the student nested functions up to the penultimate layer are denoted as $g_s(\cdot)$ and the student projection head is denoted as $f_s^p(\cdot)$. Therefore, the query representations $q_s$ and the positive keys $k_t^+$ are given by:

$$q_s = f_s^p(g_s(x_s)), \quad k_t^+ = f_t^p(g_t(x_t)), \tag{1}$$

where $g_t(\cdot)$ denotes the teacher nested functions up the penultimate layer and $f_t^p(\cdot)$ is the teacher projection head. $f_s^p$ and $f_t^p$ are two-layer perceptrons. Besides, the cached $i$-th negative key in the queue is denoted as $k_{t_i}^-$ which is produced the same way as $k_t^+$ but from the preceding batches. The fixed-size teacher dictionary $K = \{k_{t_1}^-, \cdots, k_{t_N}^-\}$ contains $N$ negative keys. The representations of the current batch are added to the queue, while the oldest representations are removed from the queue.

**The contrastive loss.** The value of the contrastive loss should be small when $q_s$ is close to $k_t^+$ and distant from $k_{t_i}^-$ in the representation space. To meet this condition, we consider the wildly-used and effective contrastive loss function: InfoNCE (Van den Oord et al., 2018):

$$\mathcal{L}_{ctr} = -\log \frac{\exp(q_s \cdot k_t^+ / \tau)}{\exp(q_s \cdot k_t^+ / \tau) + \sum_{i=1}^{N} \exp(q_s \cdot k_{t_i}^- / \tau)}, \tag{2}$$

where $\tau$ is a hyper-parameter that controls the concentration level. $N$ is the size of the teacher dictionary. $\mathcal{L}_{ctr}$ can be intuitively interpreted as the log loss of a softmax-based $(N+1)$-way classification task. In our case, we attempt to classify $q_s$ as $k_t^+$ in the scope of $\{k_t^+\} \cup \{k_{t_1}^-, k_{t_2}^-, \cdots, k_{t_N}^-\}$.

**The consistency in the teacher dictionary.** The introduction of the fixed-size teacher dictionary decouples the size of the memory bank from batch size and dataset size. The teacher dictionary can be larger than the commonly-used batch and smaller than the dataset. Therefore, we bypass huge batch, which aims at providing in-batch negative samples. Besides, we can avoid sampling inconsistent negative keys from the memory bank. The core to learning good features by contrastive learning lies in the rich and challenging negative representations. In CRD (Tian et al., 2020) and WCoRD (Chen et al., 2021b), the negative keys are momentum updated. The momentum update to the negative keys brings about two main issues: (1) the negative keys were updated only when they were last processed, and (2) the update interval for each negative key can be highly different. The two issues cause inconsistent negative keys. To provide consistent negative keys, we update the teacher projection head in a momentum manner. Since $g_t$ are frozen in the distillation framework, momentum updating the teacher projection head results in consistent negative keys. Specifically, denoting the parameters of $f_t^p$ as $\omega_t$ and those of $f_s^p$ as $\omega_s$, we update $\omega_t$ as:

$$\omega_t \leftarrow m_c \omega_t + (1 - m_c)\omega_s. \tag{3}$$

$m_c \in [0, 1]$ is a momentum coefficient which adjusts the update smoothness. Since $\omega_s$ are optimized by the training optimizer, the momentum update of $\omega_t$ makes the teacher projection head $f_t^p$ progress more smoothly than the student projection head $f_s^p$. Therefore, the difference between the teacher projection heads at different iterations can be made small. As a result, the negative keys encoded at different iterations can be consistent. Besides, the teacher dictionary itself is gradually updated. The representations of the current batch are enqueued, while the representations of the oldest batch are dequeued. This gradual replacement is beneficial for maintaining the consistency of the queue since the oldest representations are the least consistent with the current ones.

### 3.2 REPRESENTATIONS OF ONE CLASS FLOCK TOGETHER

As shown in Eq. 2, classifying $q_s$ as $k_t^+$ in the scope of $\{k_t^+, k_{t_1}^-, k_{t_2}^-, \cdots, k_{t_n}^-\}$ is a discrimination on an instance level. However, $k_{t_1}^-$ which shares the same class label with $q_s$ should be close to $q_s$ in the representation space. Simply rejecting those $k_{t_1}^-$ is not beneficial for the student learning good features. To bring $q_s$ closer to its instance-negative but class-positive keys, we introduce a slow-moving student whose nested functions up to the penultimate layer are denoted as $g_s'(\cdot)$. Specifically, the slow-moving student is implemented as a momentum moving average of the student. The slow-moving student is

also accompanied with a projection head $f'_s$, which is also updated in a momentum manner. Denoting the parameters of $g_s$ as $\theta_s$, the parameters of $g'_s$ as $\theta'_s$ and those of $f'_s$ as $w'_s$, we update $\theta'_s$ and $w'_s$ by:

$$\theta'_s \leftarrow m_r \theta'_s + (1 - m_r)\theta_s \quad w'_s \leftarrow m_r w'_s + (1 - m_r)w_s, \tag{4}$$

where $m_r \in [0, 1]$ is another momentum coefficient and $w_s$ denotes the parameters of the student projection head $f^p_s$. Therefore, the instance-negative but class-positive keys $q^+_{s-}$ can be obtain by:

$$q^+_{s-} = f'_s(g'_s(x'_s)), \qquad \triangleleft \quad \text{the instance-negative but class-positive keys} \tag{5}$$

where $x'_s$ is another view of $x$ under the random data augmentations. Instead of directly narrowing down the distance between $q_s$ and $q^+_{s-}$, we use $q_s$ to predict $q^+_{s-}$, which softens the constraint. Formally, a predictor $h_s$, implemented as a two-layer perceptron, is proposed to produce the prediction $p_s \triangleq h_s(q_s)$. The loss is simply defined as the mean squared error between $l_2$ normalized $p_s$ and $q^+_{s-}$:

$$\mathcal{L}_{pred} = \|\frac{p_s}{\|p_s\|_2} - \frac{q^+_{s-}}{\|q^+_{s-}\|_2}\|^2_2 = 2 - 2\langle \frac{p_s}{\|p_s\|_2}, \frac{q^+_{s-}}{\|q^+_{s-}\|_2}\rangle. \tag{6}$$

Furthermore, we symmetrize the loss by feeding $x'_s$ to the student and $x_s$ to the slow-moving student to compute $\widetilde{\mathcal{L}}_{pred}$. Formally, denoting the representations output from $x'_s$ by the student as $\widetilde{q}_s$ and the corresponding instance-negative but class-positive keys as $\widetilde{q}^+_{s-}$, we compute $\widetilde{\mathcal{L}}_{pred}$ by:

$$\widetilde{\mathcal{L}}_{pred} = \|\frac{\widetilde{p}_s}{\|\widetilde{p}_s\|_2} - \frac{\widetilde{q}^+_{s-}}{\|\widetilde{q}^+_{s-}\|_2}\|^2_2 = 2 - 2\langle \frac{\widetilde{p}_s}{\|\widetilde{p}_s\|_2}, \frac{\widetilde{q}^+_{s-}}{\|\widetilde{q}^+_{s-}\|_2}\rangle. \tag{7}$$

Here $\widetilde{p}_s \triangleq h_s(\widetilde{q}_s)$, $\widetilde{q}^+_{s-} \triangleq f'_s(g'_s(x_s))$ and $\widetilde{q}_s \triangleq f^p_s(g_s(x'_s))$. Note that $q^+_{s-}$ and $\widetilde{q}^+_{s-}$ are detached from the current computational graph during the distillation process.

### 3.3 TRAINING THE STUDENT

With the slow-moving student and the teacher, Eq. 2, Eq. 6 and Eq. 7 aim at assisting the student to effectively learn powerful features through contrastive learning. The student still needs to learn features from the training data. For image classification, the task-specific loss is defined as the cross-entropy loss. Overall, the total loss $\mathcal{L}_{total}$ can be formulated as:

$$\mathcal{L}_{total} = \lambda_{ctr}\mathcal{L}_{ctr} + \lambda_{pred}(\mathcal{L}_{pred} + \widetilde{\mathcal{L}}_{pred}) + \lambda_{cls}\mathcal{L}_{cls}, \tag{8}$$

where $\lambda_{ctr}$, $\lambda_{pred}$ and $\lambda_{cls}$ are three balancing factors. $\mathcal{L}_{cls} \triangleq \mathcal{H}(y, y^s)$, where $\mathcal{H}(\cdot)$ refers to the standard cross-entropy, $y$ denotes the one-hotel label and $y_s$ is the student output.

## 4 EXPERIMENTS

We validate the effectiveness of CoCoRD in improving the student performance. The student-teacher combinations are divided into two main categories: (1) students share the same architecture style with teachers, and (2) the architectures of the students are different from those of the teachers.

**Datasets.** To investigate the performance improvements of students, we employ two benchmarks: (1) CIFAR100 (Krizhevsky et al., 2009) and (2) ImageNet-1K (Russakovsky et al., 2015). CIFAR100 has 100 classes and there are 500 training images and 100 validation images per class. ImageNet-1K, a large-scale dataset, contains 1000 classes and provides 1.28 million training images and 50K validation images. To test the transferability of features that students learn by CoCoRD, we utilize two more datasets: (1) STL-10 (Coates et al., 2011) and (2) TinyImageNet (Chrabaszcz et al., 2017). We only use the 5K labeled training images and 8K validation images from 10 classes in STL-10. TinyImageNet consists of 200 classes and each has 500 training images and 50 validation images.

### 4.1 EXPERIMENTS ON CIFAR100

We experiment on CIFAR100 with 13 student-teacher combinations in total[1], 7 of which are student-teacher combinations with the same architecture style, and the remaining 6 are student-teacher combinations with different architectures. Table 1 focuses on student-teacher combinations with the same architecture style, while Table 2 provides experimental results of student-teacher combinations with different architectures. As can be observed in both tables, KD (Hinton et al., 2015), a simple yet effective method, provides a strong baseline. CoCoRD can consistently outperform KD and

---

[1]On CIFAR100, $\lambda_{ctr}=1$, $\lambda_{cls}=1$, $\lambda_{pred}=4$. More training details and data augmentations are provided in the supplementary materials

Table 1: CIFAR100 test *accuracy* (%) of students trained with different distillation methods (ours is CoCoRD), when having the same architecture style as the teacher. ↑ denotes outperforming KD, and ↓ denotes underperforming. For all the compared methods, we use author-provided or author-verified code from the CRD repository. Our reported results are the averages over 5 runs. The best result among the methods which are *not* combined with another method is shown in **bold**. The best result among the combined methods is underlined.

| Teacher
Student | WRN-40-2
WRN-16-2 | WRN-40-2
WRN-40-1 | resnet56
resnet20 | resnet110
resnet20 | resnet110
resnet32 | resnet32x4
resnet8x4 | vgg13
vgg8 |
|---|---|---|---|---|---|---|---|
| Teacher | 75.61 | 75.61 | 72.34 | 74.31 | 74.31 | 79.42 | 74.64 |
| Student | 73.26 | 71.98 | 69.06 | 69.06 | 71.14 | 72.50 | 70.36 |
| KD | 74.92 | 73.54 | 70.66 | 70.67 | 73.08 | 73.33 | 72.98 |
| FitNet | 73.58 (↓) | 72.24 (↓) | 69.21 (↓) | 68.99 (↓) | 71.06 (↓) | 73.50 (↑) | 71.02 (↓) |
| AT | 74.08 (↓) | 72.77 (↓) | 70.55 (↓) | 70.22 (↓) | 72.31 (↓) | 73.44 (↑) | 71.43 (↓) |
| SP | 73.83 (↓) | 72.43 (↓) | 69.67 (↓) | 70.04 (↓) | 72.69 (↓) | 72.94 (↓) | 72.68 (↓) |
| CC | 73.56 (↓) | 72.21 (↓) | 69.63 (↓) | 69.48 (↓) | 71.48 (↓) | 72.97 (↓) | 70.71 (↓) |
| VID | 74.11 (↓) | 73.30 (↓) | 70.38 (↓) | 70.16 (↓) | 72.61 (↓) | 73.09 (↓) | 71.23 (↓) |
| RKD | 73.35 (↓) | 72.22 (↓) | 69.61 (↓) | 69.25 (↓) | 71.82 (↓) | 71.90 (↓) | 71.48 (↓) |
| PKT | 74.54 (↓) | 73.45 (↓) | 70.34 (↓) | 70.25 (↓) | 72.61 (↓) | 73.64 (↑) | 72.88 (↓) |
| AB | 72.50 (↓) | 72.38 (↓) | 69.47 (↓) | 69.53 (↓) | 70.98 (↓) | 73.17 (↓) | 70.94 (↓) |
| FT | 73.25 (↓) | 71.59 (↓) | 69.84 (↓) | 70.22 (↓) | 72.37 (↓) | 72.86 (↓) | 70.58 (↓) |
| FSP | 72.91 (↓) | n/a | 69.95 (↓) | 70.11 (↓) | 71.89 (↓) | 72.62 (↓) | 70.23 (↓) |
| NST | 73.68 (↓) | 72.24 (↓) | 69.60 (↓) | 69.53 (↓) | 71.96 (↓) | 73.30 (↓) | 71.53 (↓) |
| CRD | **75.48** (↑) | 74.14 (↑) | 71.16 (↑) | 71.46 (↑) | 73.48 (↑) | **75.51** (↑) | 73.94 (↑) |
| LCKT | 75.22 (↑) | 74.11 (↑) | 71.14 (↑) | 71.23 (↑) | 72.32 (↑) | 74.65 (↑) | 73.50 (↑) |
| GCKT | 75.47 (↑) | 74.23 (↑) | 71.21 (↑) | 71.43 (↑) | 73.41 (↑) | 75.45 (↑) | **74.10** (↑) |
| CoCoRD (ours) | **75.48** (↑) | **75.17** (↑) | **71.74** (↑) | **72.11** (↑) | **74.20** (↑) | 75.29 (↑) | 73.99 (↑) |
| CRD+KD | 75.64 (↑) | 74.38 (↑) | 71.63 (↑) | 71.56 (↑) | 73.75 (↑) | 75.46 (↑) | 74.29 (↑) |
| WCoRD | 75.88 (↑) | 74.73 (↑) | 71.56 (↑) | 71.57 (↑) | 73.81 (↑) | 75.95 (↑) | 74.55 (↑) |
| CoCoRD+KD | 75.90 (↑) | 75.25 (↑) | 72.09 (↑) | 72.18 (↑) | 74.37 (↑) | 75.28 (↑) | 74.26 (↑) |

Table 2: CIFAR100 test *accuracy* (%) of students trained with different distillation methods, when the teachers' architectures are significantly different from those of the students. ↑ denotes outperforming KD, and ↓ denotes underperforming. Our results are the averages over 5 runs. The best result among the methods which are *not* combined with another method is shown in **bold**. The best result among the combined methods is underlined.

| Teacher
Student | vgg13
MobileNetV2 | ResNet50
MobileNetV2 | ResNet50
vgg8 | resnet32x4
ShuffleNetV1 | resnet32x4
ShuffleNetV2 | WRN-40-2
ShuffleNetV1 |
|---|---|---|---|---|---|---|
| Teacher | 74.64 | 79.34 | 79.34 | 79.42 | 79.42 | 75.61 |
| Student | 64.60 | 64.60 | 70.36 | 70.50 | 71.82 | 70.50 |
| KD | 67.37 | 67.35 | 73.81 | 74.07 | 74.45 | 74.83 |
| FitNet | 64.14 (↓) | 63.16 (↓) | 70.69 (↓) | 73.59 (↓) | 73.54 (↓) | 73.73 (↓) |
| AT | 59.40 (↓) | 58.58 (↓) | 71.84 (↓) | 71.73 (↓) | 72.73 (↓) | 73.32 (↓) |
| SP | 66.30 (↓) | 68.08 (↑) | 73.34 (↓) | 73.48 (↓) | 74.56 (↑) | 74.52 (↓) |
| CC | 64.86 (↓) | 65.43 (↓) | 70.25 (↓) | 71.14 (↓) | 71.29 (↓) | 71.38 (↓) |
| VID | 65.56 (↓) | 67.57 (↑) | 70.30 (↓) | 73.38 (↓) | 73.40 (↓) | 73.61 (↓) |
| RKD | 64.52 (↓) | 64.43 (↓) | 71.50 (↓) | 72.28 (↓) | 73.21 (↓) | 72.21 (↓) |
| PKT | 67.13 (↓) | 66.52 (↓) | 73.01 (↓) | 74.10 (↑) | 74.69 (↑) | 73.89 (↓) |
| AB | 66.06 (↓) | 67.20 (↓) | 70.65 (↓) | 73.55 (↓) | 74.31 (↓) | 73.34 (↓) |
| FT | 61.78 (↓) | 60.99 (↓) | 70.29 (↓) | 71.75 (↓) | 72.50 (↓) | 72.03 (↓) |
| NST | 58.16 (↓) | 64.96 (↓) | 71.28 (↓) | 74.12 (↑) | 74.68 (↑) | 74.89 (↑) |
| CRD | 69.73 (↑) | 69.11 (↑) | 74.30 (↑) | 75.11 (↑) | 75.65 (↑) | 76.05 (↑) |
| LCKT | 68.21 (↑) | 68.81 (↑) | 73.21 (↑) | 74.62 (↑) | 74.70 (↑) | 75.08 (↑) |
| GCKT | 68.78 (↑) | 69.20 (↑) | 74.29 (↑) | 75.18 (↑) | 75.78 (↑) | 76.13 (↑) |
| CoCoRD (ours) | **69.86** (↑) | **70.22** (↑) | **74.52** (↑) | **75.99** (↑) | **77.28** (↑) | **76.42** (↑) |
| CRD+KD | 69.94 (↑) | 69.54 (↑) | 74.58 (↑) | 75.12 (↑) | 76.05 (↑) | 76.27 (↑) |
| WCoRD | 69.47 (↑) | 70.45 (↑) | 74.86 (↑) | 75.40 (↑) | 75.96 (↑) | 76.32 (↑) |
| CoCoRD+KD | 69.26 (↑) | 69.89 (↑) | 74.62 (↑) | 76.48 (↑) | 77.39 (↑) | 76.56 (↑) |

achieve highly competitive performance compared with other state-of-the-art methods. Note that $m_c$ in Formula 3 is set to 1 for the WRN-40-2/WRN-40-1 combination. Although the teacher projection head attached to WRN-40-2 is only randomly initialized and not updated during the distillation process, CoCoRD still achieves the state-of-the-art result. This implies the features provided by the well-trained teacher from the penultimate layer are already distinguishing, which are then projected into the representation space by the frozen teacher projection head $f_t^p$.

Based on the discussion above, the teacher projection heads in Table 2 are randomly initialized since the difference in architecture style is very likely to bring about the difference in the input shape. Note

Table 3: To evaluate the transferability of features learned by different distillation methods, we employ linear probing to perform a 10-way classification on STL10 and 200-way classification on TinyImageNet. For this experiment, we use the combination of teacher WRN-40-2 and student WRN-16-2. Top-1 accuracy (%) is reported. The student baseline and teacher are trained from scratch. Details are in the supplemental materials.

| | Student | KD | AT | CRD | CRD+KD | CoCoRD | Teacher |
|---|---|---|---|---|---|---|---|
| CIFAR100→STL-10 | 69.93 | 70.82 | 70.39 | 71.36 | 71.59 | **73.63** | 68.31 |
| CIFAR100→TinyImageNet | 34.53 | 33.83 | 33.80 | 35.88 | 36.07 | **38.39** | 32.38 |

Table 4: Top-1 and Top-5 *error* rates (%) of the students ResNet-18 trained with different distillation methods on ImageNet-1K validation set. The best performance is shown in **bold**.

| | Teacher | Student | AT | KD | SP | CC | CRD | CRD+KD | ReviewKD | SSKD | WCoRD | CoCoRD |
|---|---|---|---|---|---|---|---|---|---|---|---|---|
| Top-1 | 26.70 | 30.24 | 29.30 | 29.34 | 29.38 | 30.04 | 28.83 | 28.62 | 28.39 | 28.48 | 28.51 | **28.26** |
| Top-5 | 8.58 | 10.92 | 10.00 | 10.12 | 10.20 | 10.83 | 9.87 | 9.51 | 9.49 | 9.33 | 9.84 | **9.30** |

that it is because of the projection heads that CoCoRD can achieve distillation under cross-architecture setting. The projection heads can project features at the penultimate layer of different shapes into one representation space, where we can easily define the contrastive loss based on Eq. 2.

As shown in Table 2, CoCoRD is highly effective for combinations of different architectures. Even if the teacher projection head is not updated, CoCoRD can consistently achieve the best performance compared to those not combined with another method. Especially, for the resnet-32x4/ShuffleNetV2 pair, CoCoRD presents 77.28% Top-1 accuracy, which is 1.5% higher than the second best GCKT (75.78%). On the other hand, methods based on intermediate features perform poorly with different-architecture combinations. The observation suggests that CoCoRD can largely blur the requirement for significant similarities between students and teachers. We conjecture that knowledge distillation based on features at the penultimate layer can avoid the conflicts of different inductive biases that different models exploit. This indicates that the proposed CoCoRD is more generally applicable for student-teacher combinations with different architectures.

**Limitations.** In Table 1, CoCoRD+KD does not bring further performance improvements over CoCoRD. The same phenomenon can be observed in Table 2. MobileNetV2 (Sandler et al., 2018) does not obtain more performance improvements with CoCoRD+KD. These phenomena indicate that further investigations are needed to combine CoCoRD with other knowledge distillation methods and extremely lightweight student models are still challenging for knowledge distillation.

**Linear probing.** Following CRD (Tian et al., 2020), we employ linear probing to evaluate the transferability of the student features. We freeze the student and train a linear classifier on the global average pooling features of the student to perform a 10-way classification on STL10 and 200-way classification on TinyImageNet. As shown in Table. 3, CoCoRD exhibits strong transferability and outperform the second best (CRD+KD) by a large margin on the two datasets (2.04% improvement on STL10 and 2.32% on TinyImageNet). The proposed CoCoRD, which has a negligible performance drop on CIFAR100 compared with the teacher, (Please see Table. 1), shows better transferability than the teacher (5.32% improvement on STL10 and 6.01% TinyImageNet). The linear probing experiment indicates that CoCoRD-distilled models have better generalization ability.

## 4.2 Experiments on ImageNet

To investigate the scalability of CoCoRD to large-scale datasets, we employ ResNet-18 and ResNet-34 as the student-teacher combination to perform experiments on ImageNet-1K. For a fair comparison, we follow the standard PyTorch ImageNet training practice except that we have 100 training epochs like CRD and WCoRD. We also use the PyTorch-released ResNet-34/18 as our teacher/student. On ImageNet, we set $\lambda_{ctr}$=1, $\lambda_{cls}$=1, $\lambda_{pred}$=4 and only calculate $\mathcal{L}_{pred}$. The Top-1 and Top-5 error rates of different distillation methods are provided in Table 4 (the lower, the better). The results in Table 4 show that the proposed CoCoRD achieves the best performance on the large-scale ImageNet. The relative improvement of CoCoRD over WCoRD (Chen et al., 2021b) on Top-1 error is 14.45%, and the relative improvement of CoCoRD over CRD (Tian et al., 2020) on Top-1 error is 40.43%. Both improvements validate the scalability of the proposed CoCoRD to large-scale datasets.

## 4.3 Ablation Study

### 4.3.1 Study of Encoder Combinations

By default, we use the teacher to generate representations for contrastive learning and the slow-moving student is employed to produce representations of another view of the input of the student.

Table 5: CIFAR100 test *accuracy* (%) of resnet32 trained with different encoder combinations. The best performance and the corresponding encoder combinations are shown in **bold**. The teacher is resnet110. *mean* denotes the average over 5 runs and *std* stands for the corresponding standard deviation. Note that the resnet110 is pre-trained and the resnet32 is initialized the same as the student and updated in a momentum manner.

| Option | A | | B | | C | | D | | E | | F | |
|---|---|---|---|---|---|---|---|---|---|---|---|---|
| Encoder | Contrastive **resnet110** | Cognate **resnet32** | Contrastive resnet32 | Cognate resnet32 | Contrastive resnet32 | Cognate resnet110 | Contrastive resnet110 | Cognate resnet110 | Contrastive resnet110 | Cognate - | Contrastive - | Cognate resnet32 |
| mean (±std) | **74.07** (±0.14) | | 68.56 (±0.78) | | 72.18 (±0.37) | | 73.71 (±0.34) | | 72.92 (±0.23) | | fails | |

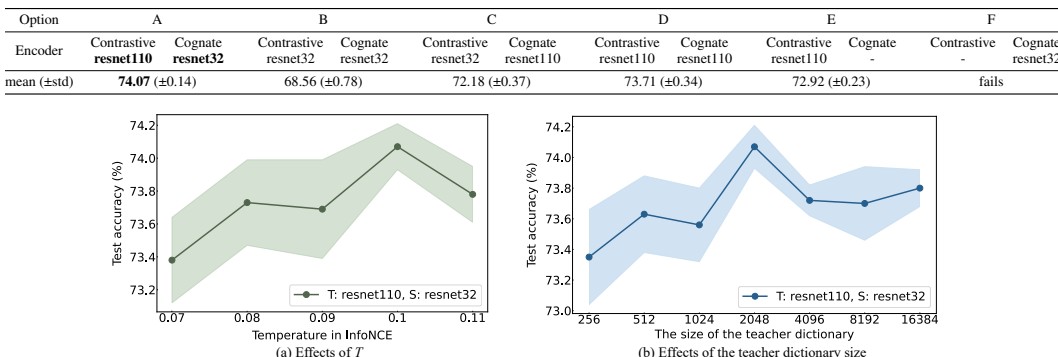

Figure 3: Effects of the temperature ($\tau$) in InfoNCE with $N$=2048, as shown in (a), and the effects of the teacher dictionary size ($N$) with $\tau$=0.1, as shown in (b).

To investigate how the representation quality affects the distillation performance, we utilize different models to provide those representations. For clarity, the model that generates dictionary-caching representations is referred to as *contrastive* encoder. The model that produces the instance-negative but class-positive representations is referred to as *cognate* encoder. Results are reported in Table 5. Comparing options A (the default option) and B, we can find that leveraging the pre-trained teacher to provide quality representations for contrastive learning is more beneficial for the distillation. Besides, removing the cognate encoder and setting $\lambda_{pred}$ to zero (option E) lead to poor performance, suggesting the cognate encoder can alleviate the adverse impact of the potential noise. If we remove the contrastive encoder and still use the dictionary with cognate encoder (option F), the distillation process fails. The results in Table 5 can support the effectiveness of each encoder in CoCoRD.

### 4.3.2 STUDY OF MOMENTUM

Table 6: CIFAR100 test *accuracy* (%) of the student resnet32 trained with different combinations of $m_c$ (in Eq. equation 3) and $m_r$ (in Eq. equation 4). The best performance and the corresponding $m_c$ and $m_r$ are shown in **bold**. The teacher is resnet110 and the accuracy of vanilla student and the teacher can be found in Table 1. *mean* denotes the average over 5 runs and *std* stands for the corresponding standard deviation.

| $m_r$ | 0 | | **0.9** | | | | | | 0 | 0.99 | 0.999 | 0.9999 | 1 | 1 |
|---|---|---|---|---|---|---|---|---|---|---|---|---|---|---|
| $m_c$ | | 0 | 0.9 | 0.99 | **0.999** | 0.9999 | 1 | | | | 0.999 | | | |
| mean | 73.00 | 72.87 | 73.34 | 73.89 | **74.07** | 73.70 | 73.80 | | 73.80 | 73.65 | 73.54 | 72.93 | 72.86 | 73.12 |
| std | 0.33 | 0.32 | 0.23 | 0.24 | 0.14 | 0.32 | 0.12 | | 0.25 | 0.22 | 0.52 | 0.40 | 0.50 | 0.20 |

As shown in Formulas 3 and 4, $m_c$ controls the progressing speed of the teacher projection head $f_t^p$, while $m_r$ manages the speed of the slow-moving student and its projection head. To investigate the impact of momentum, we employ resnet110 as the teacher to train resnet32 with different $m_c$ and $m_r$. The results are reported in Table 6. When $m_c$=$m_r$=0 and $m_c$=$m_r$=1, CoCoRD can improve the student performance. The effectiveness in both cases implies CoCoRD is robust. Besides, with $m_r$ fixed, a large value of $m_r$ (e.g. 0.99 or 0.999) works much better than $m_r$=0, suggesting that consistent representations in the teacher dictionary are beneficial for the distillation.

### 4.3.3 STUDY OF HYPER-PARAMETERS

**The temperature $\tau$.** The value of $\tau$ in Eq. 2 varies from 0.07 to 0.11. As shown in Figure 3(a), CoCoRD is sensitive to $\tau$. Both extremely high and low temperature lead to sub-optimal performance. As suggested in CRD (Tian et al., 2020), we set $\tau$ to 0.1 for experiments on CIFAR100, while $\tau$ is set to 0.07 on ImageNet. We suggest tuning the value of $\tau$ based on the classification difficulty.

**The size of the teacher dictionary.** The number of negative keys is determined by the teacher dictionary size $N$. To investigate the effects of the teacher dictionary size, we validate various values of $N$. As shown in Figure 3(b), extremely small teacher dictionary provides insufficient negative keys, leading to sub-optimal performance. However, the extremely large teacher dictionary can introduce

noise, which adversely affects the distillation performance. Based on our experiments, $N$=2048 should suffice on CIFAR100 while $N$= 65536 on ImageNet. Note that the teacher dictionary in Co-CoRD is significantly smaller than the memory banks in CRD (Tian et al., 2020) and WCoRD (Chen et al., 2021b), which is more economic for large-scale datasets.

**The balancing factors.** We conduct experiments on CIFAR100 to investigate the effects of the three balancing factors $\lambda_{ctr}$, $\lambda_{cls}$ and $\lambda_{pred}$. We use resnet32-resnet110 as the student-teacher combination. For experiments on balancing factors, we set $\tau$=0.1, $N$=2048, $m_c$=0.999 and $m_r$=0.9. "✗" denotes we set the balance factor to 0 and "✓" means we set the balance factor to the corresponding value provided in the second row. Details on simple grid search for each balancing factor can be found in the supplementary material. As we can see from Table. 7, all components in CoCoRD are essential for achieving high distillation performance. When $\lambda_{ctr}$ is set to 0, there is a serious performance drop, which indicates contrasting student representations with negative keys in teacher dictionary is necessary in improving the student performance. Moreover, by comparing the result when $\lambda_{pred}$=0 with the result when $\lambda_{pred}$=4, we can see that the slow-moving student can reduce the negative effects of the noise in the teacher dictionary.

Table 7: The effects of the three balancing factors. CIFAR100 test *accuracy* (%) is reported. The best performance is shown in **bold**. Average over 5 runs. More details can be found in the supplementary material.

| $\lambda_{cls}$ | $\lambda_{ctr}$ | $\lambda_{pred}$ | *mean (std)* |
|---|---|---|---|
| 1 | 1 | 4 | **74.20 (±0.14)** |
| ✗ | ✓ | ✓ | *fails* |
| ✓ | ✗ | ✓ | 71.81 (±0.42) |
| ✓ | ✓ | ✗ | 72.92 (±0.23) |

## 4.4 TRANSFER LEARNING

Table 8: Transfer learning. For PASCAL VOC, Faster R-CNN is fine-tuned on VOC `trainval07+12` and evaluated on `2007test`. For COCO, Mask R-CNN is fine-tuned on COCO `train2017` and evaluated on `val2017`. The Faster/Mask R-CNN models are with the R50-C4 backbones (He et al., 2017). Numbers in green indicate the performance improvement over the detectors initialized by the vanilla student. Please see the supplementary material for details. ResNet101 is the teacher with 77.37% top-1 accuracy on ImageNet.

| | Classification | Object Detection | | | | | |
|---|---|---|---|---|---|---|---|
| | ImageNet | PASCAL VOC Detection | | | CoCo Detection | | |
| | Top-1 accuracy (%) | $AP_{50}$ | AP | $AP_{75}$ | $AP_{50}$ | AP | $AP_{75}$ |
| scratch | - | 60.2 | 33.8 | 33.1 | 44.0 | 26.4 | 27.8 |
| Student | 76.15 | 81.3 | 53.5 | 58.8 | 59.9 | 40.0 | 43.1 |
| CRD | 76.86 (+0.71) | 81.7 (+0.4) | 54.2 (+0.7) | 60.0 (+1.2) | 60.5 (+0.6) | 40.7 (+0.7) | 43.9 (+0.8) |
| CoCoRD | 77.57 (+1.42) | **82.0** (+0.7) | **55.0** (+1.5) | **61.1** (+2.3) | **60.9** (+1.0) | **41.0** (+1.0) | **44.5** (+1.4) |

We further validate the feature quality of CoCoRD-distilled models by transferring the model weights to object detection task, including PASCAL VOC (Everingham et al., 2010) and COCO detection (Lin et al., 2014). We fine-tune the pre-trained models in an end-to-end manner on the target datasets. The detector for PASCAL VOC is Faster R-CNN (Ren et al., 2015) with a backbone of R50-C4. For COCO object detection, the model is Mask-RCNN (He et al., 2017) with the R50-C4 backbone. Note that the CoCoRD-distilled ResNet50 can outperform the teacher ResNet101 by 0.2% top-1 accuracy on classification. As shown in Table 8, the CoCoRD-initialized detectors exhibit better performance than the student-initialized and CRD-initialized counterparts. The valid reuse of model weights further demonstrates the transferability of CoCoRD-distilled features.

## 5 CONCLUSION

In this paper, we propose a contrastive-learning-based knowledge distillation method named Contrastive Consistent Representation Distillation. From a perspective of regarding contrastive learning as a dictionary looking-up task, we build a fixed-size dictionary to cache consistent teacher representations. Besides, to alleviate the adverse impact of the potential noise in the teacher dictionary, we employ a slow-moving student, implemented as a momentum-based moving average of the student, to provide instance-negative but class-positive targets. CoCoRD does not employ the entire dataset as the memory bank, which is economic for large-scale datasets. Extensive experiments demonstrate that CoCoRD, which utilizes fewer negative keys, can boost the performance of the students on diverse image classification datasets. Additionally, the models distilled by CoCoRD on ImageNet classification can efficiently improve object detection performance on PASCAL VOC and COCO.

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

# A APPENDIX

## A.1 QUANTITATIVE RESULTS ON THE ACHIEVED SPEED-UP, MEMORY REDUCE AND OTHERS

In the following three tables, we provide quantitative results on the achieved speed-up, memory cost reduce, and other quantitative information about the teacher/student (T/S) combinations used on CIFAR100 (in Tabs. 1 and 2) and those T/S combinations used on ImageNet (Tabs. 4 and 8). The results are measured with Intel Core i7-8700 CPU on Ubuntu 20.04 operating system and memory cost is measured by Pytorch Profiler in a forward pass.

Table 9: Quantitative results on the achieved speed-up, parameter compression and memory cost reduce. The combinations are from Tab. 1. The inference latency is measured on image of size 32x32.

| Combination(T/S) | Inference Latency (ms) | Speed-up | Memory Cost (MB) | Mult-Add | Parameters (K) |
|---|---|---|---|---|---|
| WRN-40-2/WRN-16-2 | 21.28/7.98 | 62.50% | 11.73/4.39 | 327.62M/101.12M | 2255/703 |
| WRN-40-2/WRN-40-1 | 21.28/10.51 | 50.61% | 11.73/5.87 | 327.62M/83.29M | 2255/570 |
| resnet56/resnet20 | 21.75/11.48 | 47.22% | 8.72/3.21 | 125.76M/40.82M | 862/278 |
| resnet110/resnet20 | 56.09/11.48 | 79.53% | 16.97/3.21 | 253.16M/40.82M | 1737/278 |
| resnet110/resnet32 | 56.09/17.01 | 69.67% | 16.97/5.05 | 253.16M/69.13M | 1737/473 |
| resnet32x4/resnet8x4 | 24.93/7.52 | 69.84% | 20.71/6.03 | 1.08G/177.07M | 7434/1234 |
| vgg13/vgg8 | 8.72/3.89 | 55.39% | 4.20/2.10 | 285.2M/96.33M | 9462/3965 |

Table 10: Quantitative results on the achieved speed-up, parameter compression and memory cost reduce. The combinations are from Tab. 2. The inference latency is measured on image of size 32x32.

| Combination(T/S) | Inference Latency (ms) | Speed-up | Memory Cost (MB) | Mult-Add | Parameters (K) |
|---|---|---|---|---|---|
| Vgg13/MobileNetV2 | 8.72/14.97 | - | 4.20/3.39 | 285.2M/6.54M | 9462/813 |
| ResNet50/MobileNetV2 | 17.01/14.97 | 11.99% | 3.63/3.39 | 83.67M/6.54M | 23713/813 |
| ResNet50/vgg8 | 17.01/3.89 | 77.13% | 3.63/2.10 | 83.67M/96.33M | 23713/3965 |
| resnet32x4/ShuffleNetV1 | 24.93/31.84 | - | 20.71/13.90 | 1.08G/38.72M | 7434/949 |
| resnet32x4/ShuffleNetV2 | 24.93/19.01 | 23.75% | 20.71/9.01 | 1.08G/44.52M | 7434/1356 |
| WRN-40-2/ShuffleNetV1 | 21.28/31.84 | - | 11.73/13.90 | 327.62M/38.72M | 2255/949 |

Table 11: Quantitative results on the achieved speed-up, parameter compression and memory cost reduce. The combinations are from Tabs. 4 and 8. The inference latency is measured on image of size 224x224.

| Combination(T/S) | Inference Latency (ms) | Speed-up | Memory Cost (MB) | Mult-Add | Parameters (M) |
|---|---|---|---|---|---|
| ResNet34/ResNet18 | 43.97/28.52 | 35.14% | 59.82/39.75 | 3.66G/1.81G | 21.80/11.69 |
| ResNet101/ResNet50 | 104.37/50.84 | 51.29% | 259.72/177.83 | 7.80G/4.09G | 44.55/25.56 |

Additionally, we compare the size of the teacher dictionary in the proposed CoCoRD with the size of the memory banks in CRD. Note that the keys in CRD memory banks are only 128-d while the keys in the proposed CoCoRD teacher dictionary are 2048-d. Even with higher dimensions of the stored keys, CoCoRD are still more storage efficient.

Table 12: Comparison on the size of memory bank(s). Note that there is one teacher dictionary in the proposed CoCoRD while there are two memory banks in CRD.

| | CRD | CoCoRD | Relative Size |
|---|---|---|---|
| CIFAR100 | 51.20MB | 16.78MB | 32.77% |
| ImageNet | 1311.92MB | 536.87MB | 40.92% |

## A.2 THEORETICAL STUDY

Given two deep neural networks, a teacher $f^T$ and a student $f^S$, and let $x$ be the network input. We denote representations at the penultimate layer as $f^T(x)$ and $f^S(x)$, respectively. We would like to bring $f^S(x_i)$ and $f^T(x_i)$ close while pushing apart $f^S(x_i)$ and $f^T(x_j)$ ($x_i$ and $x_j$ represent different training samples).

For clear notation, we define variables $S$ and $T$ for the student representations and the teacher ones of the data, respectively: $x \sim p_{(}x)$; $S = f^S(x)$; $T = f^T(x)$.

Let us define a distribution $q$ with variable $C$. The latent variable $C$ decides whether the tuple $(f^S(x_i), f^T(x_j))$ is drawn form the joint distribution $p(T, S)$ (when $C$=1) or drawn from the product of marginal distributions $p(S)p(T)$ (when $C$=0).

$$q(T, S|C = 1) = p(T, S), q(T, S|C = 0) = p(T)p(S)$$

Suppose we are given 1 congruent pair drawn from the joint distribution (i.e. the same input provided to T and S) for every N incongruent pairs drawn from the product of marginals (independent randomly inputs provided to T and S). Then the priors on the latent C are:

$$q(C = 1) = \frac{1}{N + 1}, q(C = 0) = \frac{N}{N + 1}.$$

By Bayes' rule and simple manipulations, the posterior for $C = 1$ is given by:

$$q(C = 1|T, S) = \frac{q(T, S|C = 1)q(C = 1)}{q(T, S|C = 0)q(C = 0) + q(T, S|C = 1)q(C = 1)} = \frac{p(T, S)}{p(T, S) + Np(T)p(S)}.$$

We can observe a connection with mutual information:

$$\log q(C = 1|T, S) = -\log(1 + N\frac{p(T)p(S)}{p(T, S)}) \leq -\log(N) + \log \frac{p(T, S)}{p(T)p(S)}.$$

Taking expectation on both sides w.r.t $p(T, S)$ and rearranging gives us:

$$I(T; S) \geq \log(N) + \mathbb{E}_{q(T, S|C=1)} \log q(C = 1|T, S),$$

where $I(T; S)$ is the mutual information between the distributions of the teacher and student representations. Though we do not know the true distribution $q(C = 1|T, S)$, a neural network can be used to estimate whether a pair comes from the joint distribution or the marginals.

By maximizing KL divergence between the joint distribution $p(T, S)$ and the product of marginal distributions $p(T)p(S)$, we can maximize the mutual information between the student representations and the teacher representations.

