# OpenReview forum: "Contrastive Consistent Representation Distillation"
_ICLR.cc/2023/Conference — Submitted to ICLR 2023_

### Official Review · Reviewer_FUQY · 2022-10-25

**Confidence:** 3
**Clarity, Quality, Novelty And Reproducibility:** 1. The paper is clearly written.
2. T…
**Correctness:** 3
**Technical Novelty And Significance:** 3
**Empirical Novelty And Significance:** 2
**Recommendation:** 6

**Details Of Ethics Concerns:**

No ethic concerns.

**Strength And Weaknesses:**

Strength:

1. The proposed method is well-motivated and novel. Improving the memory efficiency in contrastive learning is a valid practical concern.
2. The experiment results on various models and tasks show clear gains, which demonstrate the effectiveness of the method.
3. The paper provides ablation studies on all proposed components.

Weakness:

1. Can the authors provide a quantitative result on the achieved speed-up and reduce memory cost?



**Summary Of The Paper:**

The paper proposed a new contrastive distillation method that distills consistent representations with a light cost. Specifically, the proposed method uses 1) a fixed-size memory bank as the teacher dictionary to achieve the storage efficiency; 2) the moving average of the student's projection head as the teacher's projection head to produce consistent negative keys; 3) the moving average of the student model to bring $q_r$ closer to its instance-negative but class-positive keys.

The authors verify the effectiveness of the method on extensive tasks and models.

**Summary Of The Review:**

The paper proposed a new contrastive distillation method that distills consistent representations with a light cost. The topic is important and the method is supported by extensive experiments.

---

> ### Author Response · Authors · 2022-11-11
> **Quantitative results on the achieved speed-up, memory reduce and others**
>
> Dear Reviewer FUQY,
>
> Thank you very much for the constructive comments.
>
> In the following three tables, we provide quantitative results on the achieved speed-up, memory cost reduce, and other information about the teacher/student (T/S) combinations that are used on CIFAR100 (in Tabs. 1 and 2) and those T/S combinations used on ImageNet (Tabs. 4 and 8). The results are measured with Intel Core i7-8700 CPU on Ubuntu 20.04 operating system. As we can see, the proposed CoCoRD can significantly reduce the computational burden, especially on the challenging ImageNet.
>
> The following table includes the combinations in Tab. 1 on CIFAR100
> | Combination(T/S) | Inference Latency (one 32x32 image) | Achieved Speed-up | Memory Cost (Forward pass) | Mult-Add| Number of Parameters |
> |:---:|:---:|:---:|:---:|:---:|---|
> |  WRN-40-2/WRN-16-2 | 21.28ms/7.98ms |62.50% |11.73MB/4.39MB | 327.62M/101.12M| 2255K/703K |
> |  WRN-40-2/WRN-40-1| 21.28ms/10.51ms  | 50.61% | 11.73MB/5.87MB | 327.62M/83.29M| 2255K/570K|
> | resnet56/resnet20| 21.75ms/11.48ms | 47.22%| 8.72MB/3.21MB | 125.76M/40.82M| 862K/278K|
> | resnet110/resnet20| 56.09ms/11.48ms  | 79.53% | 16.97MB/3.21MB | 253.16M/40.82M| 1737K/278K|
> | resnet110/resnet32| 56.09ms/17.01ms |69.67% | 16.97MB/5.05MB| 253.16M/69.13M|1737K/473K|
> | resnet32x4/resnet8x4 | 24.93ms/7.52ms  | 69.84% |20.71MB/6.03MB |1.08G/177.07M|7434K/1234K|
> | vgg13/vgg8 | 8.72ms/3.89ms |55.39% |4.20MB/2.10MB| 285.2M/96.33M| 9462K/3965K |
>
> ***
> ***
>
> The following table provides the combinations in Tab. 2 on CIFAR100
> | Combination(T/S) | Inference Latency (one 32x32 image) | Achieved Speed-up | Memory Cost (Forward pass) | Mult-Add| Number of Parameters |
> |:---:|:---:|:---:|:---:|:---:|---|
> | Vgg13/MobileNetV2| 8.72ms/14.97ms | - |4.20MB/3.39MB | 285.2M/6.54M| 9462K/813K |
> | ResNet50/MobileNetV2|17.01ms/14.97ms | 11.99% | 3.63MB/3.39MB | 83.67M/6.54M| 23713K/813K|
> | ResNet50/vgg8|17.01ms/3.89ms|77.13%| 3.63MB/2.10MB | 83.67M/96.33M|23713K/3965K|
> | resnet32x4/ShuffleNetV1| 24.93ms/31.84ms |-| 20.71MB/13.90MB | 1.08G/38.72M| 7434K/949K|
> | resnet32x4/ShuffleNetV2| 24.93ms/19.01ms|23.75%| 20.71MB/9.01MB| 1.08G/44.52M|7434K/1356K|
> | WRN-40-2/ShuffleNetV1 | 21.28ms/31.84ms |-|11.73MB/13.90MB |327.62M/38.72M|2255K/949K|
>
> ***
> ***
>
> The following table includes the combinations in Tabs. 4 and 8 on ImageNet
> | Combination(T/S) | Inference Latency (one 224x224 image) | Achieved Speed-up | Memory Cost (Forward pass) | Mult-Add| Number of Parameters |
> |:---:|:---:|:---:|:---:|:---:|---|
> |ResNet34/ResNet18| 43.97ms/28.52ms|35.14%| 59.82MB/39.75MB| 3.66G/1.81G | 21.80M/11.69M|
> |ResNet101/ResNet50|104.37ms/50.84ms|51.29% |259.72MB/177.83MB| 7.80G/4.09G | 44.55M/25.56M |
>
> Note that the CoCoRD-distilled ResNet50 can outperform ResNet101, as shown in Tab. 8.
>
> Please don’t hesitate to let us know for any further feedback. Thanks!

---

> ### Author Response · Authors · 2022-11-11
> **Comparison on the size of the memory bank**
>
> Additionally, we compare the size of the teacher dictionary in the proposed CoCoRD with the size of the memory banks in CRD.
>
> |  | CIFAR100 | ImageNet |
> |:---:|:---:|:---:|
> | CRD | 51.20MB | 1311.92MB |
> | CoCoRD | 16.78MB | 536.87MB |
> | Relative Size | 32.77% | 40.92% |
>
> Note that the keys in CRD memory banks are only 128-d while the keys in the proposed CoCoRD teacher dictionary are 2048-d. Even with higher dimensions of the stored keys, CoCoRD are still more storage efficient.

---

> ### Author Response · Authors · 2022-12-12
> **A Gentle Reminder**
>
> Dear Reviewer FUQY ,
>
> Thanks a lot for your valuable and helpful comments. As we are approaching the end of the discussion period, could you please help to check if our response has addressed your concerns? Thank you! We are more than happy to clarify and discuss more if you have additional concerns.
>
> Best Regards,
>
> Authors of Paper 2497

---

### Official Review · Reviewer_mvUm · 2022-10-26

**Confidence:** 5
**Correctness:** 3
**Technical Novelty And Significance:** 2
**Empirical Novelty And Significance:** 2
**Recommendation:** 5

**Clarity, Quality, Novelty And Reproducibility:**

The paper is quite clear in the issues raised and their proposed fix to the issues. I also think the authors have done a good job covering different kind of experiments, e.g., image classification/transfer learning, even if I feel the results themselves are not that impressive.

The authors have included sufficient details for someone to reproduce the experiments.


**Strength And Weaknesses:**

Strengths

- The issue about memory usage is a valid point to raise. CRD [1] does indeed need a big memory bank to store the features of the whole training dataset, an issue which will get worse with a bigger dataset (e.g., ImageNet). This becomes particularly important when someone has to save the model and resume training; one does not only need to store the teacher/student’s weights, but also need to store the whole training data’s feature set.

- The other main issue, which is about the negatives potentially being out of touch with the fresh features that the teacher is computing, could be important as well. I use the phrase “potentially” because it is not clear how big of an issue that really is in practice.

- The paper is well written, the issues clearly explained, and the mathematical formulation is easy to follow.


Weaknesses

- Even though the issues that the authors have raised pertaining to CRD are valid, in the sense that it would be nice if those did not exist, those are not the issues which I feel stand out too much. As I said in the “Strengths” section, it is not clear how much effect it will have that the negatives sampled for a query were updated a while ago.

- The reason the concern raised above (the importance of recency in feature bank) is important is because the results, especially on the most comprehensive evaluation setup CIFAR-100, don’t show that big of an improvement over the existing contrastive distillation setups like CRD. Sometimes they are even worse.

- But, I think it is not just that the issue of “recency in feature banks” (in CRD) was not important enough, and hence there wasn’t that much improvement brought about by the proposed method. I think that the new proposed way of sampling negatives could potentially be not so ideal in certain situations. Here is how: the batch size for CIFAR-100 setup is set to 64 (as per supplementary) while the queue size is set to 2048. This means that a particular batch of features coming out of the teacher will persist in the queue for (2048/64) 32 iterations. This means that for many different query images, there will be the same set of negatives being used (32 times) until they finally get out of the queue. Compare this to CRD, where for each query image, the negatives will be randomly coming from the whole training set, thereby reducing the possibility of that issue. I do not know how important this difference is in practice, but the fact that there is not a consistent increase in performance for CIFAR-100 might hint at a problem like this. In summary, the method that has been proposed is not a superset of CRD in terms of (theoretical) effectiveness.

- Overall, there is not a straightforward way to fix the issues that the authors have raised about CRD. They need to introduce an additional student network to counter some issues because of the queue data structure, which leads to additional loss components which need to be tuned.


**Summary Of The Paper:**

The paper presents a way of performing knowledge distillation using contrastive learning. The authors point out that existing distillation methods that use contrastive learning have two main issues: (i) memory intensive - as it needs to store the representations of the whole dataset to construct negatives, (ii) the negatives which exist in the memory bank might not have been updated in a while, thereby potentially making the task of differentiating the positives (which are computed fresh in each iteration) from negatives (in the bank) easier than it should be. To this end, the authors propose using a queue data structure instead for storing the negatives. This queue is updated with fresh representations from the teacher and can also be much smaller than the size of the whole training dataset. Empirically, this alternative way of performing contrastive distillation helps in improving the performance of the distilled student a bit, and also helps in certain transfer learning setups.


**Summary Of The Review:**

I like that the issues pertaining to contrastive distillation methods were raised, issues which I had not thought of before, but as explained in the “Weaknesses” section, I do not feel those issues are that important, particularly when resolutions to those issues could themselves introduce other issues. I do not think that the idea produced is novel/general enough for it to be applicable to all the problem settings, as is depicted in the results as well (Table1/2). Because of these reasons, I do not recommend acceptance.

---

> ### Author Response · Authors · 2022-11-11
> **Response to Reviewer mvUm**
>
> Dear Reviewer mvUm,
>
> Thank you for the very constructive comments. Below is our response.
>
> **Response to Weakness 1**:
> We think it has non-trivial effects that the negatives were updated a while ago and will prevent the student from learning good features. If the negatives sampled for a query are updated a while ago, the negatives are not well-represented in the representation space. The distance between the anchor and the negatives may be much larger than it should be. Also, the student may not have to bring the anchor close to the positive as it should. Therefore, the student can easily find a low-loss solution with out-of-date negative representations and be prevented from learning good features. We can see the results provided in Tab. 3 where we evaluate the feature transferability. The proposed CoCoRD can outperform CRD by a large margin (+2.27% on STL-10 and +2.51% on TinyImageNet). With the same teacher/student combination (WRN-40-2/WRN-16-2), CoCoRD shows the same result as CRD on CIFAR-100 (in Tab. 1). Based on the results of classification (Tab. 1) and linear probing (Tab. 3), we can see that CoCoRD-distilled models learn better features and have better generalization ability.
>
> Besides, we think the issue of "recency in memory bank" is just a side effect of updating memory banks in a momentum manner and sampling from them. More importantly, the inconsistency in the memory bank is the bigger issue. The representation in the student memory bank was updated when it was last seen, so the sampled student representations are about the encoder (student) at multiple different steps. In other words, the sampled representations are from very different encoders over the past epoch and thus are less consistent. Taking the weighted sum of the freshly computed representation and the corresponding one in the memory bank (momentum update) does not address the inconsistency. Even worse, simply taking the weighted sum can result in representations that are less meaningful and less consistent. Those representations will be cached in the memory bank for sampling, which is not beneficial for contrastive learning. The representations in the teacher memory bank also face the issue of inconsistency because of the momentum update (simply taking the weighted sum of representations of different views). Besides, in practice, the initial values of the memory bank are random numbers like those in CRD. Updating the memory bank in a momentum manner will introduce the initial value (noise) into the keys.
>
> In the proposed CoCoRD, we only maintain one teacher dictionary where all the cached keys are negative samples. Besides, we momentum update the encoder (teacher projection head) to produce consistent negative keys. The momentum update makes the teacher projection head evolve more smoothly than the student projection head. The difference between the teacher projection heads at different steps can be made small and thus the negative keys encoded at different steps can be kept consistent as much as possible. Besides, the teacher dictionary is implemented as a queue, with the current batch enqueued and the oldest batch dequeued, avoiding introducing the meaningless noise into the keys.
>
> **Response to Weakness 2**:
> The experiments on CIFAR-100 can be divided into two main categories: (1) students share the same architecture style with teachers, and (2) the architectures of the students are different from those of the teachers. In both categories, CoCoRD only uses 2048 negative samples. The number of negative samples in CoCoRD is ***significantly smaller*** than the one used in CRD (16384 for reporting the accuracy of CRD-distilled models). As we can see from Tab. 1 (experiments on combinations of the same architecture style), CoCoRD provides highly competitive results compared with CRD. Moreover, as shown in Tab. 2 (experiments on combinations of different architectures), CoCoRD can consistently outperform CRD with unadvantageous conditions (a smaller number of negative samples and only randomly-initialized teacher projection head). This indicates that the proposed CoCoRD can largely blur the requirement for significant similarities between students and teachers, and is applicable for student-teacher combinations with different architectures. Besides, on the large-scale ImageNet, when the number of negatives in CoCoRD (65536) is the same as the one in CRD (the memory consumption of CoCoRD queue is still much less than that of CRD memory banks), CoCoRD achieves the best results and outperforms CRD by a large margin (the relative improvement of CoCoRD over CRD on Top-1 accuracy is 40.43%). When evaluating the feature quality (linear probing in Tab. 3 and object detection in Tab. 8), CoCoRD also outperforms CRD by a large margin, suggesting CoCoRD-distilled models learn better features and show stronger generalization ability. CoCoRD-distilled ResNet-50 (77.57% Top-1 accuracy) can outperform the teacher ResNet-101 (77.37% Top-1).

---

> > ### Author Response · Authors · 2022-11-11
> > **Response to Reviewer mvUm (part 2)**
> >
> > **Response to Weakness 3**:
> > The issue of "recency in memory bank" is a side effect of sampling from memory banks and performing a momentum update on the memory banks. We think the bigger issue is the inconsistent representations caused by the momentum update on the memory bank. In CRD, the negative keys are momentum-updated. The momentum update on the negative keys brings out two issues: (1) the negative keys were momentum-updated only when they were last processed, and (2) the update interval for each negative key can be highly different. The two issues cause inconsistent negatives.
> >
> > For the first "updating the representation when it was seen" issue, the representations in the memory bank are about the encoder at multiple different steps. Since the encoder is optimized aggressively, the representations generated at different steps can be inconsistent (the difference among the encoders at multiple different steps is large, thus the difference among the generated representations is large). Momentum-updating the old representations with the newly-computed ones still makes them inconsistent. The training samples cannot be well-represented by the inconsistent representations, which prevents the student from learning good features.
> >
> > For the second "update interval" ("recency in memory banks") issue, we think it is a side effect but may make the sampled negative keys more inconsistent because some sampled negatives may be momentum updated but some are not updated for many steps (the difference is enlarged).
> >
> > In the proposed CoCoRD, we only maintain one small teacher dictionary (a queue). ***The momentum update is adopted on the encoder*** (the teacher projection head). The momentum update makes the teacher projection head evolve more smoothly than the student projection head. Although the keys are still encoded by encoders at different steps, the difference among these encoders can be made small. Therefore, the encoded keys are kept as consistent as possible. With consistent representations provided, good features can be learned by a number of well-represented and challenging negative samples.
> >
> > Regarding the example given in the comment, we think the queue helps keep the teacher dictionary consistent. With the batch size set to 64 and the queue size set to 2048, the most "up-to-date" keys will not be more than 32 (2048/64) steps away from the oldest keys. In CRD, the negatives will be randomly sampled from the whole training set. There is no guarantee on the number of training steps between the latest representations and the oldest one (the sampled keys can be quite inconsistent).
> >
> > To sum up, the core of CoCoRD is trying to provide consistent negatives to facilitate contrastive learning. Although CoCoRD may underperform CRD in some teacher/student combinations in Tab. 1, CoCoRD can improve the memory efficiency in contrastive learning (only one teacher dictionary and much fewer negative keys: 2048 (CoCoRD) v.s. 16384 (CRD)). Besides, with students and teachers from different architectures (Tab. 2), CoCoRD can consistently outperform CRD by a large margin (16.84% average relative improvement over CRD). The experimental results on linear probing and transfer learning can also demonstrate the effectiveness of CoCoRD.
> >
> > **Response to Weakness 4**:
> > The additional student (the slow-moving student) is proposed to minimize the intra-class distance since the class-positive keys are taken as instance-negative ones in the queue. As we can see in Tab. 7, the distillation performance drops dramatically (from 74.20% to 72.92%) when we remove the slow-moving student and set $L_{pred}$ to 0. This suggests the slow-moving student can alleviate the impact of the potential noise in the queue. If we refer to the results in Tab. 2 in the supplementary materials, we can see the slow-moving student can work reasonably well across a range of $\lambda_{pred}$. The tuning of the loss is not so troublesome. Overall, we think CoCoRD is towards making the combination of contrastive learning and distillation more efficient. We hope knowledge distillation can enjoy more benefits from contrastive learning or self-supervised learning.
> >
> > **Response to Summary of the Review**:
> > Only considering a method itself (the third portion in Tabs. 1 and 2), although CoCoRD may not consistently achieve the best results in the similar-architecture settings (Tab. 1), it outperforms all others in the different-architecture settings (Tab. 2), suggesting CoCoRD is applicable for combinations with different architectures. We think the different-architecture settings are more general and practical because we may not always have the similar-architecture teachers. CoCoRD can largely blur the requirement for significant similarities between students and teachers. Besides, the CoCoRD-distilled models have better feature transferability (shown in Tabs. 3 and 8). The effectiveness of CoCoRD on large-scale ImageNet (Tab. 4) also indicates the proposed method is general.

---

> > > ### Author Response · Authors · 2022-11-12
> > > **Welcome any further feedback**
> > >
> > > Please don’t hesitate to let us know for any further feedback. Thanks!

---

> > ### Comment · Reviewer_mvUm · 2022-12-07
> > **Response to the authors**
> >
> > I thank the authors for their very detailed response. However, my major concerns still stand. I am not disagreeing that the negative representations cannot become inconsistent with the positive, because of which the task can become much easier for the student than it needs to be. My point is that we do not know how important that issue is, and whether that is really what happens. One way to show this could be to compare the distance of student's representation to the negatives sampled from teacher in two different cases: (i) in the vanilla CRD (baseline) and (ii) the proposed method, after a fixed number of iterations. According to the hypothesis put forward by the authors, we should see that the distances in (i) are larger (and hence the student does not need to do much) compared to (ii). A result like that will at least quantitatively show that there is a signal which can be used in a training process like the authors have done.
> >
> > However, unless the experiment described above shows us something really surprising, for example, the original CRD paper was optimizing an already dead loss function (because the student is already different from teacher's negatives), I do not think the contribution of this work is that useful/general purpose. I hence keep my initial rating.

---

> > > ### Author Response · Authors · 2022-12-08
> > > **Response to Reviewer mvUm**
> > >
> > > Dear Reviewer mvUm:
> > >
> > > Thank you for your feedback. We intend to address your major concerns from the following two aspects.
> > >
> > > **(1)** With all due respect, we do not attempt to make the negatives consistent/inconsistent with the student representation. In fact, **we make the negative keys consistent at different steps by momentum-updating the encoder** (make the difference of one representation at different steps small). The distance between the negatives and the representation from the student should be large because that is what contrastive learning does. However, the inconsistency in the memory bank leads to badly-represented negatives and could make the distance between the student representation and the negatives inaccurate in the representation space (much larger or smaller than it is supposed to be), which keeps the student from learning good features with contrastive learning.
> > >
> > > In the proposed method, the momentum update is adopted on the encoder (the teacher projection head). With the current batch enqueued and the oldest batch dequeued, the encoded keys in the queue (the teacher dictionary) are kept as consistent as possible.
> > >
> > > To provide quantitative results, we calculate the linear centered kernel alignment (CKA) similarity between one negative key and the rest in the memory bank, and we average the CKA similarity over the entire memory bank. The memory bank is updated by the respective update strategies after 5 epochs. The results are shown as follows.
> > >
> > > |      Method             | average CKA | standard deviation |
> > > | --------------------- | ----------- | ------------------ |
> > > | CRD                   | 0.3058      | 0.0516             |
> > > | CoCoRD (ours)  | 0.1876      | 0.0226             |
> > >
> > > As we can see, the average CKA in the teacher dictionary of the proposed CoCoRD is much smaller than that of CRD, indicating negative keys are more discriminative from each other, which is more beneficial for contrastive learning with InfoNCE loss.
> > >
> > > **(2)** We only use the slow-moving student to minimize the intra-class distance since the class-positive keys are taken as instance-negative ones in contrastive learning (noise caused by the teacher queue). As we can see in Tab. 7, the distillation performance drops dramatically (from 74.20% to 72.92%) when we remove the slow-moving student and set $L_{pred}$ to 0. This suggests the slow-moving student can alleviate the impact of the potential noise in the queue.
> > >
> > > Please don’t hesitate to let us know for any further feedback. Thanks!

---

> > ### Author Response · Authors · 2022-12-12
> > **Have your concerns been addressed?**
> >
> > Dear Reviewer mvUm,
> >
> > Thanks a lot for your valuable and helpful comments. Could you please help to check if our response has addressed your concerns? Thank you! We are more than happy to clarify and discuss more if you have additional concerns.
> >
> > Best Regards,
> >
> > Authors of Paper 2497

---

### Official Review · Reviewer_gYcu · 2022-10-30

**Confidence:** 3
**Correctness:** 3
**Technical Novelty And Significance:** 2
**Empirical Novelty And Significance:** 2
**Recommendation:** 5

**Clarity, Quality, Novelty And Reproducibility:**

- Clarity: Overall, the paper is clear and easy to read, with a few exceptions, particularly with respect to section 3.2 and the notations used.
- Quality & Novelty: The quality in terms of the presentation is good. As mentioned above the novelty is limited with respect to the proposed methodology.
- Reproducibility: The procedure of the proposed method is adequately described. The code was not provided as part of the supplementary.


**Strength And Weaknesses:**

**Strengths**

1. The proposed method is evaluated on several sets of vision tasks including classification, transferability, and object detection. The performance of the method looks promising with highly competitive results.
2. The effectiveness of the proposed method is shown in the experiments and ablations. The paper also contains comparisons with relevant baselines from recent years.
3. The paper generally reads well and is clear for the most part.

**Weaknesses**

1. While the results seem good, I find the novelty in this work limited. Most of the ideas (like momentum update of params) already exist and are known in the self-supervised learning and contrastive learning literature (though these ideas have not been applied in knowledge distillation).

2. The motivation for section 3.2 is unclear. Mainly, the choice of using a slow-moving student model is not motivated or discussed. Furthermore, it is unclear how using a new view of the same input image ensures $q_r$ to become closer to "instance-negative but class-positive keys".

Other Minor comments:

1. I find the notation to be a little unclear. In equation 1, what does the index $r$ in $q_r$ refer to? It has not been introduced before. Is it supposed to be $s$ instead, since the input to the student model was $x_s$?
2. It was mentioned in the introduction that when the representations are inconsistent, "the student can easily contrast positive and negative samples", why is that?
3. In the introduction, Page 1 (last para, 3rd line): "The student representations in the memory" shouldn't it be "teacher representations" instead?


**Summary Of The Paper:**

This paper introduces and mitigates some challenges encountered in contrastive representation distillation. In particular, the authors point out that the existing methods (CRD and wCoRD) keep track of a memory bank for the negatives that often contain inconsistent representations, which can adversely affect the student model performance. To that end, the paper proposes using separate heads for the teacher and student models where the teacher head is updated using the student head using momentum, allowing the teacher representations to be consistent across different iterations. As a side benefit, the proposed model requires a smaller teacher dictionary than previous works. The paper also proposes reducing the euclidean distance between samples referred to as "instance-negative class positive" samples using an additional slowly moving student model. Several experiments show highly competitive results in teacher-student settings.


**Summary Of The Review:**

I think the motivation of the paper is sound and the proposed method combines several existing techniques to address the challenges in contrastive representation distillation. While the experimental analysis looks promising, I think the novelty is limited.

---

> ### Author Response · Authors · 2022-11-11
> **Response to Reviewer gYcu**
>
> Dear Reviewer gYcu
>
> Thank you for the very constructive comments. Below is our response.
>
> **Response to Weakness 1**:
> The technical methods in the proposed CoCoRD indeed already exist and are well-known in contrastive learning. Leveraging those methods may seem to be less ground-breaking. However, improving the memory efficiency in contrastive representation distillation is also a valid practical concern. With the existing methods, achieving consistent representations distillation that has a light cost is also exalting. CRD and WCoRD both maintain a student memory bank and a teacher one. Each of the two memory banks stores the representations of the whole training dataset. There will be a very troublesome memory usage issue with large-scale datasets, such as ImageNet. In CoCoRD, we only maintain one teacher memory bank on-the-fly, which is significantly smaller than the entire training dataset and facilitates contrastive learning. We also employ a "slow-moving" student to alleviate the impact of the potential noise in the teacher memory bank. Besides, introducing ideas from contrastive learning literature into knowledge distillation can forge a connection between the two communities. The connection can encourage us to leverage strong methods from contrastive learning to significantly improve the SOTA in knowledge distillation.
>
> **Response to Weakness 2**:
> The motivation for using a slow-moving student is to alleviate the adverse impact of the potential noise in the teacher dictionary, as we state in Page 2 (the second paragraph, the 17th line). Specifically, employing a slow-moving student is to minimize the intra-class distance. With the InfoNCE loss, the contrastive learning through the teacher dictionary is to draw distinctions on instance level. The cached teacher representations which share the same class label with the student ones are mistakenly treated as negative samples. The student representations will be pushed away from those teacher representations, enlarging the intra-class distance (however, they should be close in the representation space since they belong to the same class for the main task). Simply rejecting those instance-negative but class-positive representations is not beneficial for the student learning good features. $L_{pred}$ is computed between representations generated from $x_s$ and $x_s^{\prime}$ which are two different views of the same input image $x$. Minimizing the distance in the representation space between $x_s$ and $x_s^{\prime}$, which obviously share the same class label, can be a simple yet effective way to minimize the intra-class distance and is an alternative way of bringing $q_r$ closer to "instance-negative but class-positive keys".
>
> **Response to Minor Comment 1**:
> We are sorry for the confusion caused by the notation in Eq. 1. We intended to use $q_r$ to represent the anchor query representation produced by the student that is updated by the optimizer. Since there are two student networks: one is updated by the optimizer and the other one is in momentum update, we thought that using $q_r$ will facilitate using $q_{r_-}^+$ to represent the instance-negative but class-positive keys which are from the slow-moving student. However, we understand the confusion introduced by the unclear notation and we will revise the notation in the revision. Thank you for pointing that out.
>
> **Response to Minor Comment 2**:
> To symmetrize the distillation loss, CRD or WCoRD use the teacher key as the anchor to choose one positive and several negative samples over the student memory bank. When sampling the representations from the student memory bank, the sampled student representations are essentially about the student at multiple different steps. Besides, the student representation of a sample in the student memory bank was momentum-updated when the sample was last processed. This leads to the inconsistency in the student memory bank we mentioned. This issue may lead to trivial solutions and is not beneficial for contrastive learning. When using the teacher representations to contrast through the inconsistent student representations, the student can easily distinguish the positive sample from the negative samples. In other words, the student can easily find a low-loss solution but is prevented from learning good representations.
>
> **Response to Minor Comment 3**:
> There is a student memory bank in CRD and WCoRD (two distillation methods that are based on contrastive learning). The student representations in the student memory bank are inconsistent because the momentum-updated student representations will distinctly differ from those not updated in that step. In the proposed CoCoRD, we only maintain one small teacher dictionary. Therefore, we can avoid sampling inconsistent student representations by discarding the student memory bank.
>
> **About reproducibility**:
> The code will be made publicly available.
>
> Please don’t hesitate to let us know for any further feedback. Thanks!

---

> ### Author Response · Authors · 2022-12-12
> **Your feedback is important to us**
>
> Dear Reviewer gYcu,
>
> Thanks a lot for your constructive comments. As we are approaching the end of the discussion period, we would like to know whether our response have addressed your initial concerns about our work. We are more than happy to clarify and discuss more if you have additional concerns.
>
> Best Regards,
>
> Authors of Paper 2497

---

### Official Review · Reviewer_iNks · 2022-11-02

**Confidence:** 4
**Clarity, Quality, Novelty And Reproducibility:** Moderate Quality, Moderate Novelty, B…
**Correctness:** 2
**Technical Novelty And Significance:** 2
**Empirical Novelty And Significance:** 2
**Recommendation:** 3

**Strength And Weaknesses:**

Strength:
1. Sufficient experiments have been conducted on both classification and detection.
2. Good paper writing.

Weakness:
1. The performance improvements are not very significant. As claimed by the authors in the limitation section, a combination of their method with the previous KD does not lead to good performance.
2. Lack of in-depth study on how the inconsistency of representations in the memory bank leads to negative influence. It will be better if quantitative results can be provided to study the relation between inconsistency and the performance of knowledge distillation.
3. Lack of theoretical or experimental study on why does their method works.
4. Some factors in their framework do not have enough ablation studies. For example, what if the branch of the slow-moving student is not used?
5. SOTA knowledge distillation method usually distill the knowledge in not only the final layer but also the intermediate layers. However, in the proposed method, it seems that a projection head is required, which is usually only utilized at the final layer in previous contrastive learning methods. So I wonder whether this method can be used in knowledge distillation on the intermediate layers.
6. In my opinion, the novelty of this paper is not that large, since the moving average trick has been widely used in contrastive learning, And KD+Contrastive Learning is also not novel.

**Summary Of The Paper:**

This paper propose a new way to combine knowledge distillation with contastive learning, which updates the encoders in a momentum manner to obtain efficient and consistent encoding. Experiments on both classification and detection have been conducted.

**Summary Of The Review:**

Please refer to the strength and the weakness.

---

> ### Author Response · Authors · 2022-11-12
> **Response to Reviewer iNks**
>
> Dear Reviewer iNks
>
> Thank you for the comments. Below is our response.
>
> **Response to Weakness 1**:
> We clarify that it’s important to see the relative improvement, rather than only focus on the absolute improvement. With much fewer negatives and only one teacher dictionary, CoCoRD still provides highly-competitive results in Tab. 1. Moreover, in Tab. 2 (experimental results of combinations with different architectures), CoCoRD can consistently outperform CRD with unadvantageous conditions (only randomly-initialized teacher projection head). This indicates that the proposed CoCoRD is applicable for student-teacher combinations with different architectures (16.84% average relative improvement over CRD). Besides, on the large-scale ImageNet dataset, when the number of negative samples in CoCoRD (65536) is the same as the one in CRD, CoCoRD achieves the best results and outperforms CRD by a large margin (40.43% the relative improvement on Top-1 accuracy). Besides, the CoCoRD-distilled models have better feature transferability (shown in Tabs. 3 and 8), indicating the proposed method is general and effective.
>
> In the limitation section, the combination of CoCoRD with the previous KD does not bring further performance improvements for every teacher/student combination. We assume the reason why the model does not obtain further improvement with CoCoRD+KD is that CoCoRD draws distinctions on an instance level while KD focuses on the class level. The two methods may conflict. This only indicates further investigations are needed when combining CoCoRD with other knowledge distillation methods. The proposed CoCoRD itself is still effective based on extensive experiments. The limitations can guide us the direction of future work.
>
> **Response to Weakness 2**:
> The negative impact of inconsistent representations can be observed in Tab. 6. For the cached negative keys, the momentum update is ***on the encoder*** (the teacher projection head, denoted as $f_t^p$) in the proposed CoCoRD, ***not*** on the negative keys. With the current batch enqueued and the oldest batch dequeued, the representations in the queue are essentially about the encoders at multiple different steps. The momentum update on $f_t^p$ in Eq. 3 makes $f_t^p$ evolve smoothly and a larger value of $m_c$ in Eq. 3 leads to a smoother progress. The difference between the encoders at different steps can be made smaller with a larger value of $m_c$. Thus, the negative keys encoded by $f_t^p$ at different steps are kept as consistent as possible despite the evolution of $f_t^p$. As shown in Tab. 6, with $m_r$ fixed, a relatively large momentum coefficient (e.g. $m_c$=0.999) works much better than a small value ($m_c$=0.9 or 0, the negative keys are less consistent), suggesting that consistent negative keys are more beneficial for the contrastive-learning-based distillation.
>
> **Response to Weakness 3**:
> The theoretical study is provided in the next section of response. Please refer to it. The experimental study shown in Tabs. 5, 6, 7, and those in Tab.1 in Supplementary Material (show the effects of projection heads or predictors) can support the effectiveness of each module in CoCoRD. Empirically, we show our method is more memory efficient and provides highly competitive resutls compared with other 15 recent distillation methods published in top conferences. Especially, the proposed CoCoRD is efficient with teacher/student combinations of different architectures. On ImageNet, the proposed CoCoRD consistently outperforms all 8 recent distillation methods.
>
> **Response to Weakness 4**:
> The result of removing the slow-moving student can be found in Tab. 5, Option E. With the slow-moving student not used, the distillation performance drops from 74.07% (Option A) to 72.92% (Option E). This indicates the slow-moving student can alleviate the negative effects of the potential noise in the teacher dictionary. Besides, if we remove the contrastive encoder and still use the dictionary with the slow-moving student (Option F in Tab. 5), the distillation process fails. If we remove the contrastive encoder and the queue, there is still a serious performance drop (from 74.20% to 71.81%, shown in Tab. 7) with only the slow-moving student, which indicates contrasting student representations with negative keys in the teacher dictionary is necessary for improving the student performance. The results in Tabs. 5, 6, 7, and those in Tab.1 in Supplementary Material (show the effects of projection heads or predictors) can support the effectiveness of each module in CoCoRD. Besides, the effects of the temperature in InfoNCE and the effects of the teacher dictionary size are provided in Fig. 3.

---

> > ### Author Response · Authors · 2022-11-12
> > **The theoretical study**
> >
> > Given two deep neural networks, a teacher $f^{T}$ and a student $f^{S}$, and let $x$ be the network input.
> > We denote representations at the penultimate layer as $f^{T}(x)$ and $f^{S}(x)$, respectively.
> > We would like to bring $f^{S}(x_i)$ and $f^{T}(x_i)$ close while pushing apart $f^{S}(x_i)$ and $f^{T}(x_j)$ ($x_i$ and $x_j$ represent different training samples).
> > ***
> > ***
> > For clear notation, we define variables $S$ and $T$ for the student representations and the teacher ones of the data, respectively: $x\sim{p_(x)}$; $S=f^{S}(x)$; $T=f^{T}(x)$.
> >
> > Let us define a distribution $q$ with variable $C$. The latent variable $C$ decides whether the tuple ($f^{S}(x_i)$, $f^{T}(x_j)$) is drawn form the joint distribution $p(T, S)$ (when $C$=1) or drawn from the product of marginal distributions $p(S)p(T)$ (when $C$=0).
> >
> > $q(T, S|C=1)=p(T, S)$, $q(T, S|C=0)=p(T)p(S)$
> > ***
> > ***
> > Suppose we are given 1 congruent pair drawn from the joint distribution (i.e. the same input provided to T and S) for every N incongruent pairs drawn from the product of marginals (independent randomly inputs provided to T and S). Then the priors on the latent C are:
> > $q(C=1)=\frac{1}{N+1}$, $q(C=0)=\frac{N}{N+1}$.
> >
> > By Bayes' rule and simple manipulations, the posterior for $C=1$ is given by:
> > $q(C=1|T, S)=\frac{q(T, S|C=1)q(C=1)}{q(T, S|C=0)q(C=0) + q(T, S|C=1)q(C=1)} = \frac{p(T, S)}{p(T, S)+Np(T)p(S)}$.
> >
> > We can observe a connection with mutual information:
> > $\log q(C=1|T, S) = -\log (1+N\frac{p(T)p(S)}{p(T, S)}) \le -\log(N)+\log\frac{p(T, S)}{p(T)p(S)}$.
> >
> > Taking expectation on both sides w.r.t $p(T, S)$ and rearranging gives us:
> >
> > $I(T; S)\ge\log(N) + \mathbb{E}_{q(T, S|C=1)} \log q(C=1|T, S)$, where $I(T; S)$ is the mutual information between the distributions of the teacher and student representations.
> >
> > Thus, maximizing $\mathbb{E}_{q(T, S|C=1)} \log q(C=1|T, S)$ w.r.t the parameters of the student network increases a lower bound on mutual information.
> >
> > By maximizing KL divergence between the joint distribution $p(T, S)$ and the product of marginal distributions $p(T)p(S)$, we can maximize the mutual information between the student representations and the teacher representations.

---

> > ### Author Response · Authors · 2022-11-12
> > **Response to Reviewer iNks (part 2)**
> >
> > **Response to Weakness 5**:
> > To the best of our knowledge, distilling knowledge at the penultimate layer has more benefits than distilling in the intermediate layers, especially for different-architecture combinations. Most of the SOTA knowledge distillation methods are based on features before the final fully-connected layer (logits), like CRD and GCKT. Even KD, a simple yet effective method that is based on the features at the penultimate layer, provides a strong baseline and can outperform many distillation methods that are based on the intermediate features, like FitNets, AT, FSP, NST, and AB. We can observe it from Tabs. 1 and 2. Choosing the transfer connections between different architectures is very important when using intermediate-feature-based methods. Besides, finding an appropriate distance matrix to measure the distance between the intermediate features is non-trivial. As shown in Tab. 2, methods based on intermediate features perform poorly with different-architecture combinations. On the other hand, with the projection heads, the proposed CoCoRD can largely blur the requirement for significant similarities between students and teachers, and the performance gain of CoCoRD in Tab. 2 is noticeable. Applying the proposed method in the intermediate layers needs further investigations and is out of the scope of this paper. The proposed method aims at providing consistent representations for an efficient and effective contrastive-learning-based distillation method with less memory burden.
> >
> > **Response to Weakness 6**:
> > Leveraging existing tricks in contrastive learning may seem less ground-breaking. However, improving the memory efficiency in contrastive representation distillation is also a valid practical concern. With the existing technical tricks, achieving consistent representations distillation that has a light cost is also exalting. This becomes particularly important when we have to save the model and resume training; one not only needs to store the model weights (teacher and student) but also needs to store the whole memory bank. The proposed CoCoRD is more memory efficient and provides highly competitive results with fewer negative keys that are kept as consistent as possible. The proposed CoCoRD can enjoy a broader usage (better feature transferability and transferring the CoCoRD-distilled model weights to object detection). Besides, introducing ideas from contrastive learning literature into knowledge distillation can forge a connection between the two communities that may evolve independently. The connection can encourage us to leverage strong methods from contrastive learning to significantly improve the SOTA in knowledge distillation.
> >
> > Please don’t hesitate to let us know for any further feedback. Thanks!

---

> ### Author Response · Authors · 2022-12-12
> **Your feedback is important to us**
>
> Dear Reviewer iNks,
>
> Thanks a lot for your valuable and helpful comments. Could you please help to check if our response has addressed your concerns? Thank you!
>
> Best regards,
>
> Authors of Paper 2497

---

### Author Response · Authors · 2022-11-18
**Welcome any further discussion**

Dear reviewers,

We have revised our manuscript and added APPENDIX section for the reviewers' comments.

Please don’t hesitate to let us know for any further feedback. Thanks!

---

### Author Response · Authors · 2022-12-12
**A Gentle Reminder**

Dear reviewers and AC,

Thank you for your valuable comments. We have revised the manuscript thoroughly with additional experimental results and discussion, as detailed in our response.

Since the time for discussion period is running out, this is just a gentle reminder for the answers and revisions. We truly appreciate your comments and look forward to your further feedback.

Sincerely,

Authors

---

### Author Response · Authors · 2022-12-13
**Anticipating your further feedback**

Thanks for all reviewers careful comments and very useful suggestions. We would be more than happy if your further feedback or discussion could be provided

---

### Decision · Program_Chairs · 2023-01-20

**Decision:**

Reject

**Justification For Why Not Higher Score:**

After discussion and consideration of the rebuttal, reviewers felt the technical methods were not sufficiently insightful over prior work, and did not find the empirical results strong enough to overweigh the limited technical novelty. Questions also remained about why the method works better than prior work, and whether it is really due to achieving better consistency between student and teacher. The AC agrees with the consensus opinion.

**Justification For Why Not Lower Score:**

N/A

**Metareview: Summary, Strengths And Weaknesses:**

Summary:
This paper improves upon contrastive representation distillation (CRD) by 1) using a smaller memory bank for the teacher, and 2) introducing several momentum-based techniques that aim to keep the teacher and student encoders in sync, avoiding the issue of stale cached features leading to poor quality negatives.

Strengths:
* Identifies and aims to improve two shortcomings of contrastive-based distillation (high memory use and stale negatives)
* Good results on several benchmarks
* Extensive experiments

Weaknesses:
* Small improvement over CRD and other prior work
* The main technical ideas -- contrastive distillation, momentum encoders -- are well known in the contrastive literature and reviewers and the AC did not find substantial new insight into these methods here